# Barriers Affecting Breastfeeding Practices of Refugee Mothers: A Critical Ethnography in Saskatchewan, Canada

**DOI:** 10.3390/ijerph21040398

**Published:** 2024-03-25

**Authors:** Shela Akbar Ali Hirani

**Affiliations:** Faculty of Nursing, University of Regina, 516 RIC, 3737 Wascana Parkway, Regina, SK S4S 0A2, Canada; shela.hirani@uregina.ca

**Keywords:** breastfeeding, barriers, Canada, refugee, mothers, health, environment, support, critical ethnography

## Abstract

Refugee mothers are vulnerable to cultural stereotyping and socioeconomic hardships when they migrate to a new country. This vulnerability often has a negative impact on refugee mothers’ breastfeeding practices. Saskatchewan is one of the growing provinces in Canada that has a noticeable increase in refugee population with young children and limited availability of healthcare settings with baby-friendly status. Considering existing gaps in knowledge, this critical ethnographic study aimed to explore barriers that impede the breastfeeding practices of refugee mothers in Saskatchewan. After seeking ethics approval, data were collected using multiple methods, including in-depth interviews undertaken with 27 refugee mothers with young children of age range 1 day to 24 months, a review of media communications and field observations of community-based services/facilities available to refugee mothers. Findings suggest that psychosocial barriers, healthcare barriers, environmental barriers, and maternal and child health-related barriers impede the breastfeeding practices of refugee mothers in Saskatchewan. Breastfeeding practices of refugee mothers can be promoted through healthcare support, culturally appropriate services, interpretation services in healthcare settings, implementation of baby-friendly initiatives, hospital and community-based breastfeeding campaigns, and follow-up services. Collaborative efforts by healthcare settings, healthcare providers, policymakers, public health agencies, service providers, and governments are essential to support the breastfeeding practices of refugee mothers.

## 1. Introduction

Refugee mothers are vulnerable to cultural stereotyping and socioeconomic hardships when they migrate to a new country [1]. This vulnerability often has a negative impact on refugee mothers’ breastfeeding practices, related to social, emotional, psychological and physical stressors [2]. Breastfeeding in a new country often places refugee women in an environment where they face contradictions and conflict [2]. Refugee mother and child dyads are especially challenged by food insecurities and the distribution of commercial complementary foods by various service providers and humanitarian agencies [3], which negatively affects the breastfeeding practices of refugee mothers.

Refugee mothers in general are reported to have a lesser duration of breastfeeding due to individual and environmental barriers [4,5], despite healthcare recommendations [6]. Refugee mothers arriving in a new country often present with language difficulties, and they may not have health insurance coverage [1]. In addition, they may lack access to culturally and gender-sensitive healthcare services with breastfeeding support in refugee camps and host countries [1,2,4]. These cumulative obstacles create barriers to the utilization of breastfeeding support services by refugee mothers [1]. The health care system of the host country, hospital policies, postpartum services, and availability of breastfeeding support in the native language also affect the breastfeeding experiences of refugee mothers in a new country [7]. Although the availability of healthcare support towards the breastfeeding of refugee mothers and the baby-friendly status of healthcare settings is reported to increase the initiation and duration of breastfeeding, only 37 health centres/health authorities in Canada are reported to have baby-friendly status [8].

Saskatchewan is one of the rapidly growing Canadian provinces that supports baby-friendly initiatives (BFI) [9]. However, despite this voiced initiative, Saskatchewan has only one healthcare facility with a baby-friendly status [9] and has a limited number of healthcare facilities providing services to refugee mothers with young children. Over the past 15 years, rapid growth in the number of refugee mothers with young children is noted in this province, which indicates an increased need for breastfeeding support for refugee mothers and suggests the need for increased inclusive BFI in all healthcare settings [10,11,12].

Previous studies undertaken with refugee mothers in Canada provide insight into challenges encountered by refugee mothers in general, including personal beliefs [13,14,15], financial hardships [16,17], inadequate support from healthcare providers [18], challenges in accessing healthcare services [19] and inconsistencies in provincial breastfeeding guidelines and policies [20]. Despite the initiation of the “Interim Federal Health Programme (IFHP)” and other provincial health insurance plans that cover basic and supplemental healthcare costs for refugees, it is essential to understand the appropriateness, relevance, and quality of the offered services to breastfeeding refugee mothers in Canada. This is especially relevant to provinces such as Saskatchewan which have a noticeable influx of refugees. To promote, protect and support the breastfeeding practices of refugee mothers in Saskatchewan and minimize potential risks towards the health of young refugee children, it is imperative to have an in-depth understanding of factors that facilitate or impede breastfeeding practices of refugee women accessing and utilizing healthcare services in Saskatchewan.

Previously undertaken studies [13,14,15,16,17,18,19,20] do not uncover barriers surrounding breastfeeding practices of refugee mothers accessing and utilizing Canadian healthcare settings. There are significant gaps in the knowledge and a lack of empirical studies that focus on the breastfeeding practices of refugee mothers accessing and utilizing healthcare services in Saskatchewan, Canada. Saskatchewan’s noticeable increase in the population of refugees with young children, its limited availability of healthcare settings with baby-friendly status, potential risks towards the health of young refugee children after breastfeeding discontinuation, and existing gaps in the knowledge, all suggest a pressing need to explore challenges to the breastfeeding practices of refugee mothers. There is a defined need for empirical research on the barriers surrounding refugee breastfeeding mothers’ access and utilization of healthcare services in Saskatchewan, as well as an exploration of recommendations from refugee mothers regarding their needs for breastfeeding support programs and interventions in the healthcare settings of Saskatchewan. This study aimed to explore the barriers that impede the breastfeeding practices of refugee mothers accessing and utilizing healthcare services in Saskatchewan, Canada.

## 2. Materials and Methods

A critical ethnographic study design was employed to examine the barriers that directly and indirectly affect the breastfeeding practices of refugee mothers accessing and utilizing healthcare services in Saskatchewan. Critical ethnography is identified as an appropriate design to undertake this research because this design provides an opportunity to critically examine the issues surrounding the lives of people experiencing struggles and vulnerability [21,22]. Moreover, this design provides an opportunity to examine the experiences of a potentially vulnerable group and analyse the association of those experiences (multiple realities) with power and truth [23]. Hence, critical ethnography helps a researcher to gain insight into the range of factors in social structures that shape the experiences of oppressed groups [21], in this case, refugee mothers of young children.

This study was undertaken in Saskatchewan, Canada. Ethical approval was sought from the University of Regina Ethics Review Board before the commencement of the data collection (number 2020-104). Data were collected from the year 2020 to 2022 using multiple methods, including in-depth interviews with refugee mothers with young children aged 1 day to 24 months, field observations of community-based services/facilities available to refugees in Saskatchewan, and a review of media communications. In view of the emancipatory agenda of critical ethnography, the use of multiple methods of data collection helped in uncovering the truth from a deeper level, in triangulating data, and in examining the range of factors that affect the breastfeeding experiences of refugee mothers.

In-depth interviews were conducted with refugee mothers having young children aged 1 day to 24 months until saturation (richness) in data was achieved. Altogether, 27 interviews were undertaken. Participants were recruited using a purposive and snowball sampling method from different cities of Saskatchewan, including Regina, Saskatoon, Prince Albert, Swift Current and Moose Jaw. Refugee mothers were recruited with the help of the refugee settlement organizations offering services to the refugee population in Saskatchewan. A semi-structured interview guide was utilized to undertake interviews of 40–60 min with refugee mothers. The questions intended to explore the breastfeeding experiences of refugee mothers and key barriers that negatively affect their breastfeeding practices. Due to the COVID-19 pandemic and related restrictions, all interviews were conducted via Zoom. Interviews were conducted in refugee mothers’ preferred languages (mainly English and Arabic). Of 27 mothers, 24 refugee mothers preferred to be interviewed in Arabic and the rest of the mothers were interviewed in English. After seeking informed consent, data were collected by the principal investigator and a research assistant fluent in English and Arabic. All interviews were audio recorded. Interviews conducted in Arabic were transcribed and translated into English by the research assistant. An audit trail of the interviews conducted in Arabic was undertaken by a language expert to check the accuracy of the translation. Field notes were maintained by the researcher during data collection.

Field observations were conducted by the researcher to observe healthcare and environmental barriers experienced by refugee mothers in Saskatchewan. The researcher made observations on services available to refugee mothers, including social support services for refugee families, availability of interpretation services, privacy to breastfeed in public, helpline services in different languages, cost of transportation services to healthcare, and hospital practices during the pandemic. Field observations assisted in identifying a range of accessibility challenges encountered by breastfeeding refugee mothers with young children.

To supplement the field observations, a review of media communications focusing on breastfeeding services and resources for refugee mothers was undertaken. The researcher reviewed websites maintained by the health authorities, refugee settlement services, public libraries and social services actively working in Saskatchewan. Altogether, four online reports and six websites maintained by governmental and non-governmental agencies in Saskatchewan were reviewed. Field notes were maintained during the document review.

The data were analysed iteratively and inductively. Data gathered through multiple methods, including in-depth interviews, field notes, media communications, and field observations were analysed manually by the researcher. Multiple steps were followed to analyse data. Firstly, we developed codes representing breastfeeding challenges encountered by refugee mothers. Approximately 60 codes were derived from the data gathered through in-depth interviews with participants and field notes gathered during the review of media communications/reports and field observations. Secondly, codes were classified (into categories) representing the root cause of breastfeeding challenges (maternal factors or external factors). As a next step, four broad themes (i.e., psychosocial barriers, healthcare barriers, environmental barriers, and maternal–child health-related barriers) were derived to present the range of barriers that directly and indirectly affect the breastfeeding practices of refugee mothers in Saskatchewan.

The trustworthiness of the data was assured by undertaking member checks with the study participants. Refugee mothers were contacted to seek verification of the interpretations from the data gathered through multiple sources. Triangulation of data was another strategy that facilitated the researcher to gain in-depth insight into the barriers negatively affecting the breastfeeding practices of refugee mothers in Saskatchewan. Throughout the process of data analysis and presentation, the anonymity and confidentiality of the study participants were assured by using identification numbers.

## 3. Results

### 3.1. Demographic Characteristics of Study Participants

The demographic characteristics are summarized in Table 1. Of the 27 refugee mothers, 8 were 35 years of age or older, 12 were 30 to 34 years old, 5 were between the ages of 29 and 21, and 2 were under the age of 20 years. Twenty-five study participants were living in the urban cities of Saskatchewan, whereas only two were residing in the smaller cities of Saskatchewan. Twenty-two refugee mothers arrived in Canada within the last five years, four mothers arrived in Canada six years ago, and only one mother arrived seven years ago. The education level of refugee mothers in this study indicate that 4 had a university or college education, 6 had a grade 12 education, and 17 had a grade 11 education or less. Twenty-four refugee mothers in this study were from the Middle East (Syria) and three participants were from Africa (Eritrea, Somalia and Sudan).

All the refugee mothers in this study were married and unemployed at the time of data collection. Eight mothers lived in households of seven or more, six lived in households of six people, nine participants lived in households of five people, and the minority of participants lived in households with four to three persons. Concerning type of family, 25 of 27 refugee mothers in this study were living in a nuclear household and only two participants were living in an extended household. The majority of refugee mothers were Muslims and only two mothers reported being Christian. Eight mothers had five or more children, six mothers had four children, two mothers had two living children, nine mothers had three living children, and two had one living child. Just over half of the refugee mothers’ youngest children were over a year old and less than two years old, nine mothers had a child between the ages of seven to twelve months old, and only four mothers had an infant six months old or younger. Seventeen mothers reported having no earning family members in the home, while seven mothers reported having one earning family member, and only three mothers reported two earning family members in their household. The frequency of visits to healthcare settings was also reported, with 18 of 27 participants reporting visiting healthcare settings at least once every 3 months, some more frequently. Additionally, five mothers reported visits to healthcare settings at least every six months, two mothers reported visiting healthcare settings as needed, and two mothers reported visiting healthcare settings at the time of their child’s vaccination. Additionally, all participants reported having healthcare insurance or coverage with their universal healthcare services or Saskatchewan healthcare services available to them through their Saskatchewan health card. When asked about breastfeeding practices, 17 of 27 mothers reported that they were currently exclusively breastfeeding or had done so in the past, with 10 mothers reporting that they were either currently breastfeeding and supplementing with formula or had done so in the past.

### 3.2. Barriers Affecting Breastfeeding Practices of Refugee Mothers

Findings suggest that psychosocial barriers, healthcare barriers, environmental barriers, and maternal and child health-related barriers lead to physiological challenges, mental health issues, and trauma that negatively affect the breastfeeding practices of refugee mothers. Each of these themes is discussed below.

#### 3.2.1. Psychosocial Barriers

All of the refugee mothers living in the nuclear family shared that, in the host country, they have limited contact with the extended family members who were their primary source of informational, tangible and emotional support. Mothers who gave birth to their elder child in their home country shared that, previously, they availed breastfeeding support from their extended family members, as these family members used to provide them with need-based breastfeeding guidance, psychological support, help in meal preparation, and management of household chores/child care. Mothers shared that, in this new country, as breastfeeding mothers they often miss the advice and emotional support from their extended family members. While sharing these feelings, one of the participants who found it challenging to sustain her breastfeeding practices shared,

As a refugee we always miss our family to be around us to emotionally support us. For me, I was tired psychologically more than physically. I needed psychological support so much

Study participants further shared that, in their host country, they feel isolated and emotionally deprived as they do not have anyone who can talk to them, offer breastfeeding support, provide help with household chores, and babysit their children in case they have to leave their home to seek a job or attend a doctor appointment. Refugee mothers who had children under 10 years (who require adult supervision and cannot be left unattended) shared that it is stressful to visit healthcare settings during the time of their pregnancy, childbirth, postnatal period, and/or during an event of emergency health condition as they have no extended family or free of cost babysitting facilities. Mothers shared that these stressors often negatively affect their breastfeeding practices.

#### 3.2.2. Healthcare Barriers Affecting Breastfeeding

Refugee mothers spoke about many healthcare barriers that directly and indirectly affected their breastfeeding. These barriers include hospital violation of breastfeeding guidelines, inadequate breastfeeding support from the healthcare providers, poor quality mother–baby care in hospitals, inadequate/lack of culturally appropriate food in hospitals, lack of cultural sensitivity, non-availability of interpretation services, and racism in the healthcare settings.

Many mothers shared that they were offered formula milk after their child’s birth with an assumption that they would formula feed their babies. Many mothers shared that they received formula samples in hospitals and health clinics. Field observations also support that, when asked, many health clinics in Saskatchewan offered formula samples during the infant’s clinic visit. Study participants shared that, soon after childbirth, if mothers face issues with their breastfeeding, they receive formula instead of breastfeeding counselling from the healthcare professionals. Most of the study participants further shared that, instead of formula milk they wish they were provided with prenatal or postnatal education, lactation counselling and need-based support on breastfeeding from the healthcare providers soon after their child’s birth. With great sorrow a refugee mother shared the following:

In the hospital, they [healthcare providers] asked me do you want to breastfeed or give him [infant] the formula? I said breastfeeding. But no one gave me any good idea about why to breastfeed or if is it good to breastfeed…They [healthcare providers] offer formula if mothers don’t breastfeed or don’t have enough milk.

Refugee mothers shared that, things affecting their breastfeeding practices include incompetent healthcare professionals and their lack of breastfeeding knowledge, the inconsistent care among different healthcare professionals, the lack of breastfeeding rooms/privacy in healthcare settings, the long wait times in emergency rooms, the inadequate postnatal care in hospitals, and the cases of early discharge and physical separation from their infant soon after childbirth and/or in an event when they are sick and hospitalized. A refugee mother shared the following:

After my C-section [childbirth] I stayed in hospital for one day (24 hrs). The nurse told me that they have a shortage of room and staff, and I am doing good so don’t need any more care. She told me that my son [newborn] would stay a long time in the hospital and I could go home to feel well but could come and see my son. My husband saw my situation, I wasn’t able to even walk and he decided to rent me a room in a hostel in the hospital that was paid for by my husband. It was winter time and I couldn’t move properly at that time so I stayed ten days in the hostel. In the hospital, they [healthcare professionals] used to let me breastfeed him for ten minutes only at the beginning and after two hours they would give him another type of milk [formula milk]…sometimes the nurses used to confuse me by giving me different instructions, some would encourage me to breastfeed my baby, however, others never encouraged me to breastfeed.

Refugee mothers shared that the non-availability of interpretation services and lack of access to information on breastfeeding in their preferred language negatively affect their breastfeeding practices. A review of the health authority’s website and field observation suggested that English was the main medium of service provision and interpretation services were not available in all healthcare settings. Although the health authority’s website has information on the benefits of breastfeeding, refugee mothers were unable to access breastfeeding resources available online due to a lack of guidance on the availability of these resources, inadequate English language skills, inadequate computer skills, and non-availability of a digital device. Study participants further shared that only a few healthcare settings in Saskatchewan have an availability of interpretation services; however, the interpreters are not professionally trained and are unable to interpret accurately on the breastfeeding challenges mothers are encountering, hence this causes an add-on barrier to seeking need-based breastfeeding support in healthcare settings.

Almost all of the participants shared that, during their postnatal stay in government-funded hospitals, the meal size was too small and that meals were not culturally appropriate. One of the participants, who was Muslim and could not sustain her breastfeeding practices during postnatal time, shared that the lack of culturally appropriate food in the hospital increased stress for herself and her spouse, who had to bring food from home. The participant shared the following:

Being a breastfeeding mother, I couldn’t eat the food that was provided in the hospital…the food wasn’t halal. In the beginning, they [healthcare providers] asked me to write my dietary restrictions. I told them I do not eat pork…no meat or chicken for me. They didn’t offer halal food. My husband and my friends bought me food. I only ate the breakfast in hospital.

Most of the study participants who wanted to sustain their breastfeeding practices but faced barriers, spoke about the racism and unsupportive behaviour of healthcare professionals, which caused them embarrassment and trauma while they availed of healthcare services as breastfeeding mothers. A refugee mother shared the following: 

Some nurses were very welcoming but others were not very friendly and they would be unhappy if I asked them questions. Sometimes I used to take my elder daughter with me because her English is better than mine. One nurse refused to answer my daughter’s question and told my daughter it is not your business to know, I have told your mother the situation and if she didn’t understand it, it is not my problem, I have suffered at the beginning. I was so sad and embarrassed.

Another participant who could not sustain her breastfeeding practices due to emotional distress and inadequate support from the healthcare professionals shared the following:

The doctor was racist. When I felt unwell, I was scared to tell the doctor that I was tired just to avoid encountering verbal abuse and racism. One day, I left the doctor’s office crying. I felt hurt. I was new to Canada; I was far from home and family. Her [doctor’s] words made me too sad. Health professionals should treat people all the same without racism or discrimination.

#### 3.2.3. Environmental Barriers

Refugee mothers spoke about various environmental barriers that negatively affect their breastfeeding practices, mainly inaccessibility to healthcare settings due to lack of transportation. Additionally, field observations supported the notion that cold weather conditions in Saskatchewan (cold up to −50 °C) affected the access to healthcare of many refugee families with young children when using public transportation. One of the refugee mothers whose baby was born prematurely and faced multiple challenges accessing the healthcare system shared the following:

Here in Canada, the absence of sunlight and cold cause bone pain. I am a newcomer here and don’t know how to find a doctor. Transportation is not an easy thing. Ambulance service is expensive if you don’t have private insurance. My husband would drop me off and go to work and sometimes I had to take a taxi to go home. I was told that the hospital would pay me for the taxi but I had to pay for my transportation to the hospital.

Although healthcare services are funded by the provincial government, the use of ambulance services in Saskatchewan are costly. Refugee mothers have had to pay up to CAD 400 for pick up from their home to the hospital. A review of the cost sheet on ambulance services in Saskatchewan supported this data and suggested a basic pick-up rate of $325 for Saskatchewan residents who have valid health cards and $360 for non-residents of this province, with additional escort fees if needed. A review of media communication suggested a much higher cost of CAD 465 for the use of air ambulance services for Saskatchewan residents. This cost does not include ground transportation or ambulance services to and from the airport.

Many refugee mothers in this study who had more than one child of less than 12 years of age shared that they were unable to visit healthcare settings with their younger child as they could not leave their children unattended at home. The field observations supported that, in healthcare settings, there are no babysitting options available for the patients. Additionally, community-based daycare facilities are limited and have no short-term babysitting options. Refugee mothers wished to have affordable and accessible child daycare facilities in their neighbourhood or healthcare settings where they could keep their children during their healthcare visits.

Lack of privacy to breastfeed in healthcare settings and public places was another environmental barrier that negatively affected the breastfeeding practices of refugee mothers. The field observations supported the notion that breastfeeding in many public places in Saskatchewan is not well supported and that businesses do not have any breastfeeding rooms.

Many refugee mothers who were using veils and preferred to cover themselves chose to breastfeed their babies in washrooms, neglect their child’s hunger in public places or feed baby formula that was detrimental to the child’s health and wellbeing. Refugee mothers belonging to diverse cultural backgrounds highlighted the importance of adequate privacy to breastfeeding in the hospital and in public to be able to sustain their breastfeeding practices.

Breastfeeding outdoors is a problem, especially when I go to the health clinic, I always worry about how I will breastfeed my baby over there. I often tell myself if my baby takes the bottle, it would be easier for me and better because of no privacy in public places.

The COVID-19 pandemic was an additional environmental barrier that negatively affected the breastfeeding practices of many refugee mothers. Refugee mothers who had sick babies and were admitted to the neonatal intensive care units faced physical separation from their babies, which affected their breastfeeding practices. Many new mothers who gave birth to babies during the COVID-19 pandemic ended up with complications like breast engorgement and breast abscess due to restrictions in healthcare services, lack of direct contact with healthcare professionals, no community-based follow-up visits by the nurses, and disruption in lactation counselling services. The field observations also supported the notion that, during the COVID-19 pandemic, it was difficult to get doctor’s appointments, helplines were busy, easy-to-understand breastfeeding resources were not available, and breastfeeding counselling services were paused for a longer period. Many refugee mothers who already struggled with the language barrier in their new country faced additional breastfeeding challenges due to the COVID-19 pandemic. One of the participants shared the following:

COVID changed a lot of things. During COVID-19 nurses could not visit me at home. They called me twice but I did not understand what they were saying, even my daughter [grade nine student] did not understand what they were saying. It was a difficult time.

Mothers who had a lack of access to healthcare support during the pandemic felt neglected, helpless and depressed. Refugee mothers recommended the need for follow-up services in their languages and accessible breastfeeding counselling facilities during the pandemic so that they could seek need-based support and guidance surrounding their breastfeeding practices.

#### 3.2.4. Maternal and Child Health-Related Barriers

The maternal and child health issue was another barrier that increased stress and impeded the breastfeeding practices of refugee mothers. Refugee women who had a negative experience during labour and childbirth, C-section delivery, prolonged delivery, or any complications before, during or after childbirth faced challenges in sustaining their breastfeeding practices. Moreover, mothers whose babies were born prematurely, had any health issue or disability, and required hospitalization faced additional challenges in sustaining their breastfeeding practices due to maternal and child separation. Refugee mothers who had previously faced the trauma of displacement found it challenging to cope with the additional stressors related to their own or their child’s health. 

A participant, who gave birth to a premature baby, required surgery and had an elder daughter who got sick during the time of her hospitalization, spoke about the multiple stressors affecting her breastfeeding practices. She shared the following: 

The pressure we live under here in Canada is much more than we can handle. If my oldest kids were young, it would have been way harder on me. The pain after the surgery, my premature baby, my sick daughter and other responsibilities all were so much to handle at the same time.

Another participant whose child was born with a disability and subsequently faced many challenges surrounding breastfeeding shared the following:

My daughter was born with a disability. I suffered a lot after I gave birth to her because she was born with a disability… I was depressed.

Refugee mothers shared that these stressors would have been lowered if they had social support and adequate breastfeeding support from healthcare providers in healthcare settings. Refugee mothers whose babies were sick, preterm and hospitalized and who faced breast engorgement and mastitis wished that healthcare providers in the neonatal care intensive unit would have taught them how to express and store breastmilk. They felt that this support would have sustained their breastfeeding practices and prevented them from complications.

## 4. Discussion

Refugee women often deal with multiple stresses linked to both motherhood and displacement [24]. Promoting, protecting and supporting the breastfeeding practices of refugee mothers is vital to minimizing potential risks regarding the health of young refugee children. This study identified the role of psychosocial barriers, healthcare barriers, environmental barriers, and maternal and child health-related barriers in impeding the breastfeeding practices of refugee mothers accessing and utilizing healthcare services in Saskatchewan. This study provided preliminary data that will be useful in informing healthcare providers regarding the breastfeeding practices of refugee mothers in Saskatchewan, Canada.

This study underscores that psychosocial barriers and lack of social support in a host country are the key barriers to breastfeeding practices of refugee mothers. Refugee mothers, who experience family separation and forced migration from one country to another, are at risk of experiencing reduced physical, mental and emotional wellbeing, especially during the perinatal phase of their lives [1,25]. Refugee mothers require support to maintain their traditional breastfeeding practices as these practices promote infant health, growth and development. This has been indicated by studies which indicate that breastfed infants are less susceptible to wasting and low weights [4] and are more resistant to infection [26] during displacement and resettlement in a new country. Feeding breastmilk to refugee children who are in high stress living situations reduces the risk of mortality and supports healthy growth and development [27].

Healthcare barriers form another set of challenges that impede the breastfeeding practices of refugee mothers and which negatively affects the health of young refugee children due to the use of a formula that holds more risks than benefits. These healthcare barriers suggest the need for adequate breastfeeding support from healthcare professionals, culturally sensitive care (interpretation services, cultural food in hospitals and healthcare free from racism), and implementation of baby-friendly initiatives in healthcare settings in Saskatchewan. Five challenges associated with migration that affect the ability of new mothers to breastfeed include “language barriers, racism, discrimination, poverty and separation from culture and family, and separation from their culture” [28] (p. 51). Many refugee mothers who have been torn from their home countries due to war and conflict experience anxiety and insecurity that interferes with lactation [29]. To improve the accessibility of breastfeeding services to refugee mothers, this study suggests the importance of offering prenatal and postnatal breastfeeding counselling services and educational materials to refugee mothers in multiple languages. As language barriers and the non-availability of interpretation services in healthcare settings are common barriers for refugees, which negatively affect their accessibility to healthcare [30], the use of paid professional interpreters in the community and health is vital in supporting and empowering new refugee mothers [31]. The International Society for Social Pediatrics and Child Health (ISSOP) Migration Working Group stressed that unmet prenatal, perinatal and postnatal needs, often related to language barriers, may lead to increased mother and infant morbidity and mortality [32]. Healthcare services offered in their language have been shown to expedite healthcare access by refugee women [28]. A study conducted with refugee women in Finland indicated that interpreters reduce obstacles to quality care throughout all three of the following phases of maternity care: deciding to seek care, identifying and reaching the health facility, and receiving adequate and appropriate care [31].

Adequate nutrient intake is crucial for maintaining maternal energy balance, and suboptimal nutrition can affect breast physiology and milk production [33]. The findings suggest that, for refugee mothers, who are already experiencing stress due to relocation, the absence of culturally approved foods in hospitals provides yet another impediment to the establishment of successful breastfeeding. This study further suggests the negative effects of racism on the breastfeeding practices of refugee women. The recent growth of the population of refugee mothers in Saskatchewan appears to be associated with the unintended cultural and institutional or structural displays of racism that hinder successful breastfeeding. Racism is thought to negatively affect health through a variety of complicated processes, including the harmful physiological reactions to stress that can occur through both individual and systemic pathways [34]. Women of colour are reported to have lower rates of breastfeeding continuation when compared with white women [35], this suggests the crucial need to overcome racism and address breastfeeding inequities in the healthcare system. Cultural racism, or the influence of negative stereotypes upon care providers, can be reduced through education programs for the healthcare providers that describe the cultural and psychosocial needs of refugee mothers with young children.

This study further highlighted the negative effects of environmental factors on the breastfeeding practices of refugee mothers, mainly lack of transportation, cold weather conditions, lack of child daycare facilities, lack of privacy to breastfeed, and the COVID-19 pandemic. The resettlement context is reported to influence refugee families’ access to healthcare services in the host countries [30]. Studies undertaken with the vulnerable and marginalized group of mothers highlight a variety of barriers that affect their breastfeeding practices. These barriers include lack of transportation [36], non-availability of affordable child daycare facilities in communities [37], weather conditions [38], lack of privacy in public spaces [39], and the COVID-19 pandemic [40,41]. Considering the range of environmental factors negatively affecting the breastfeeding practices of refugee mothers in Saskatchewan, this study suggests that refugee mothers must be offered community-based follow-up care and online breastfeeding counselling services in multiple languages. Moreover, refugee mothers must be offered culturally sensitive accommodations, support and privacy to breastfeed in healthcare settings, public parks, airports, businesses and public transportation. These services require interdisciplinary collaboration between the government, healthcare system, social services and refugee settlement agencies.

Maternal and child health status is another important factor that positively or negatively affects the breastfeeding practices of refugee mothers. The findings suggest that refugee mothers who often experience trauma due to disasters and subsequent migration face additional breastfeeding challenges related to their own or their child’s health. In general, breastfeeding challenges, such as perceived inadequate milk supply and difficulties with breastfeeding techniques, are significant contributors to breastfeeding cessation [42]. Therefore, promoting maternity care practices that support breastfeeding, such as ensuring skin-to-skin contact after birth, encouraging early initiation, and supporting cue-based feeding, is essential [43]. The method of childbirth, whether vaginal or caesarean, also affects breastfeeding initiation, where vaginal birth allows for earlier skin-to-skin contact, facilitating breastfeeding. Physical recovery and emotional adjustments during the postpartum period are critical for refugee mothers’ comfort and ability to breastfeed successfully. Positive breastfeeding experiences are associated with mothers perceiving more time for motherhood, linking to better mental health outcomes. Conversely, negative experiences arise from factors like separation from newborns, breastfeeding struggles, and perceived lack of support, leading to worse mental health outcomes [41,44]. Infants’ health, in addition to maternity, can also influence breastfeeding. Physical challenges like difficulty latching or weak sucking reflex [45] and medical conditions such as colic, tongue tie, or cleft palate [46,47] can impact effective breastfeeding, affecting latch and milk extraction. The findings suggest the importance of offering need-based healthcare support, culturally sensitive care and breastfeeding counselling services to refugee mothers in their native language. Refugee mothers who are at risk of experiencing post-traumatic stress disorder must be involved as partners in designing breastfeeding programs and services. Furthermore, healthcare professionals must be trained to conduct safety assessments, comprehend non-verbal cues, and offer psychosocial interventions to safeguard the health and well-being of breastfeeding refugee mothers [48].

This research identified gaps existing in the breastfeeding services and programs presently offered to refugee mothers in Saskatchewan. The knowledge from this project will assist in ascertaining the need for the development of baby-friendly support measures that promote quality care for breastfeeding refugee mothers in Saskatchewan. Recommendations from this study can guide the development of comprehensive, need-based, culturally sensitive and context-specific breastfeeding support interventions for refugee mothers with young children. Stakeholders from health, policy and social services in Saskatchewan can utilize the recommendations from this study in designing and developing baby-friendly programs and guidelines that can promote, protect and support the breastfeeding practices of refugee mothers. Future studies in this area must target the development and testing of community-based culturally sensitive breastfeeding support interventions for refugee mothers in collaboration with patient partners, health authorities, social institutions, governmental agencies and refugee settlement agencies.

## 5. Conclusions

Refugee mothers with young children are at risk of discontinuing their breastfeeding practices. This critical ethnographic study, undertaken with refugee mothers, identified that psychosocial barriers, healthcare barriers, environmental barriers, and maternal and child health-related barriers impede the breastfeeding practices of refugee mothers accessing and utilizing healthcare services in Saskatchewan, Canada. The findings suggest that these barriers lead to physiological challenges, mental health issues and trauma, which negatively affect the breastfeeding practices of refugee mothers. Breastfeeding practices of refugee mothers can be promoted through healthcare support, culturally appropriate services, interpretation services in healthcare settings, implementation of baby-friendly initiatives, hospital and community-based breastfeeding campaigns, and follow-up services. Collaborative efforts by healthcare settings, healthcare providers, policymakers, public health agencies, community-based service providers, and the provincial and federal governments are vital for the promotion, protection, and support of the breastfeeding practices of refugee mothers. Future research must involve patient partners, health authorities, social institutions, governmental agencies and refugee settlement agencies in the co-development of baby-friendly initiatives and breastfeeding support interventions that can promote, protect and support the breastfeeding practices of refugee mothers in Saskatchewan, Canada.

## Figures and Tables

**Table 1 ijerph-21-00398-t001:** Demographic characteristics of study participants.

Characteristics	Findings
Refugee mother’s age	
>35 years 30–34 years 21–29 years <20 years	8 12 5 2
Residence within Saskatchewan	
Urban cities Small cities	25 2
Arrival in Canada	
Within 5 years More than 5 years ago	22 5
Country of origin	
Middle East (Syria) Africa (Eritrea, Somalia and Sudan)	24 3
Education level	
University/college education Grade 12 education Grade 11 or less	4 6 17
Number of people in household	
7 or more 6 people 5 people 3–4 people	8 6 9 4
Type of family	
Nuclear family Extended family	25 2
Religion	
Muslim Christian	25 2
Number of children	
5 or more children 4 children 3 children 2 children 1 child	8 6 9 2 2
Number of earning members in family	
No earning member 1 earning member 2 earning members	17 7 3
Frequency of visits to healthcare settings	
At least once every 3 months At least once every 6 months As needed At the time of child’s vaccination	18 5 2 2
Feeding method	
Exclusive breastfeeding Mixed feeding	17 10

## Data Availability

The data contains sensitive information about study participants and based on the informed consent process data containing participant information cannot be shared publicly.

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
