# Peer review of "Barriers Affecting Breastfeeding Practices of Refugee Mothers: A Critical Ethnography in Saskatchewan, Canada"

_ijerph, 2024, doi:10.3390/ijerph21040398_

Round 1
Reviewer 1 Report
Comments and Suggestions for Authors
This is a most important topic. Your manuscript would be stronger if it was made clear when the data were collected. Was the timeframe immediately after COVID? In the concluding remarks it would be helpful to indicate what future research recommendations you have. What are your next steps in this program of research?
Author Response
Response to the Reviewer 1: Thanks for the appreciation and valuable feedback. All suggested changes are incorporated in the revised draft of this manuscript.
Comment 1: Your manuscript would be stronger if it was made clear when the data were collected. Was the timeframe immediately after COVID?
Response 1: Timeframe of the study is now added [Refer to line 100 and 116].
Comment 2: In the concluding remarks it would be helpful to indicate what future research recommendations you have. What are your next steps in this program of research?
Response 2: Future recommendations and next steps are added in the conclusion as well [Refer to line 540-544].
Reviewer 2 Report
Comments and Suggestions for Authors
Dear Author (s)
Thank you for this timely and informative article. I am happy to write back to you that the article is well written and if you could attend to the minor corrections, then this could be a good piece of publication.
• The article responds to the question on the barriers that directly and indirectly affect the breastfeeding practices of refugee mothers accessing and utilizing healthcare services in Saskatchewan.
• The author has utilized critical ethnographic study design to help address the broader research question. In the methods, the data collection methods are adequate including field observations which are part of critical ethnographic design.
• I find the findings original based on the methods that the author applied. This is the most relevant section in this article as mothers give their primary account through narratives, the author also conducted field observations which are later triangulated with other research methods to provide a tight interpretation of the findings. The design itself fills the gap by giving the refugee mothers a voice on what barriers exist and how they affect breastfeeding practices specifically in Saskatchewan, Canada.
• One main new knowledge in this study is that based on the voices of the refugee mothers, it recommends the development of baby-friendly support measures that will promote quality care for breastfeeding refugee mothers in Saskatchewan and this can be used to develop a culturally sensitive and context-specific breastfeeding support interventions for refugee mothers with young children.
• Since this is a qualitative study, I suggest no further controls the methodology is fine and data collection methods are adequate.
I am glad that the author (s) provides a key take-home policy message that emanates from the findings. The conclusion advocates for collaborative efforts for promoting and the establishment of breastfeeding zones that are context-specific and culturally sensitive by various stakeholders in the healthcare setting such as healthcare policymakers, public health agencies, community-based service providers, and the provincial and federal governments. These actors are also important in promoting, protecting and supporting the breastfeeding practices of refugee mothers.
• The conclusion is thus adequate.
• References are appropriate
• No additional comments on tables or figures.
Kindly attend to the following minor corrections:
- In lines 38-39, do those cultural and gender-sensitive breast feeding support exist in refugee camps? Let this come out clearly.
-Line 88 the authors need to cite the "previous studies that did not uncover barriers surrounding breastfeeding practices...."
Author Response
Response to Reviewer 2:
Thanks for the appreciation and valuable feedback. All suggested changes are incorporated in the revised draft of this manuscript.
Comment 1: In lines 38-39, do those cultural and gender-sensitive breastfeeding support exist in refugee camps? Let this come out clearly
Response: The challenge of non-availability of culturally and gender-sensitive breastfeeding support is now highlighted [Refer to line 40-41].
Comment 2: Line 88 the authors need to cite the "previous studies that did not uncover barriers surrounding breastfeeding practices...."
Response: Citation for previous studies is added [Refer to line 72].
Reviewer 3 Report
Comments and Suggestions for Authors
It is a great pleasure to review your interesting paper, I have some questions and concerns. Here is my concerns:
Abbreviation before used should be explained by full words. “Canadian provinces that support BFI” What does BFI mean?
Methods:
“Critical ethnography is identified as an ap-106 propriate design to undertake this research because this design provides an opportunity 107 to critically examine the issues surrounding the lives of people experiencing struggles and 108 vulnerability [21,22]. Moreover, this design provides an opportunity to examine the expe-109 riences of a potentially vulnerable group and analyze the association of those experiences 110 (multiple realities) with power and truth [23]. Hence, critical ethnography helps a re-111 searcher to gain insight into the range of factors in social structures that shape the experi-112 ences of oppressed groups [21], in this case, refugee mothers of young children.”
These sentences are not related to method and should be moved to the introduction section. In addition, the introduction section is too long, please summarize it.
This sentence is repetitive
"The critical ethnographic study was undertaken in Saskatchewan, Canada"
Please omit all repetitive words of “Saskatchewan, Canada". Once is enough.
In qualitative studies there is no sample, please change “sample” word to participants in this sentence: A sample of 27 123 refugee mothers with young children aged 1 day to 24-month-old was recruited using a 124 purposive and snowball sampling method “
In-depth interviews were conducted with refugee mothers in English. This sentence is not correct based on the sentences below:
Most of the refugee mothers preferred to be interviewed in Arabic. 134 After seeking informed consent, data was collected by the principal investigator and a 135 research assistant fluent in English and Arabic. Interviews were done in Arabic and then were translated to English?
Were the texts retranslated from English to Arabic? Were the texts again given to the women to confirm the translated texts from English to Arabic?
Is the language of all Arab countries the same? Do they have different dialects?
Why was the purposeful method not used for the study? For example, based on a quota, the number of years of asylum? Reason for asylum? Motivation for pregnancy? The number of mother's children? Mother's education level? Mother's place of residence? Mother's job and income? Aren't all these things very important in selecting people for interviews?
How did the data reach saturation?
How was the access to the participants? “
To supplement the field observations, a review of media communications ……” How were they chosen?
Result section: Please design a table and put your data on it.
Please write about creating themes How many codes were obtained?
How were the codes classified?
Where were the mothers' countries of origin?
Please indicate which mother the quotes belong to.
Author Response
Response to the Reviewer 3
Thanks for the appreciation and valuable feedback. Suggested changes are incorporated in the revised draft of this manuscript.
Comment 1: Abbreviation before used should be explained by full words. “Canadian provinces that support BFI” What does BFI mean?
Response: Full form of BFI is added [Refer to line 50-51].
Comment 2: Methods: “Critical ethnography is identified as an … mothers of young children.” These sentences are not related to method and should be moved to the introduction section. In addition, the introduction section is too long, please summarize it.
Response: Critical ethnography is a study design. As per journal’s guidelines it was placed in the methods section rather than in the introduction. As suggested, introduction is summarized.
Comment 3: This sentence is repetitive "The critical ethnographic study was undertaken in Saskatchewan, Canada" Please omit all repetitive words of “Saskatchewan, Canada". Once is enough.
Response: As suggested, above mentioned receptivity phrases are omitted and kept wherever they are required to clarify the context of the presented information.
Comment 4: In qualitative studies there is no sample, please change “sample” word to participants in this sentence: A sample of 27 123 refugee mothers with young children aged 1 day to 24-month-old was recruited using a purposive and snowball sampling method”
Response: Word ‘sample’ is removed from the above-mentioned sentence.
Comment 5: In-depth interviews were conducted with refugee mothers in English. This sentence is not correct based on the sentences below: Most of the refugee mothers preferred to be interviewed in Arabic. 134 After seeking informed consent, data was collected by the principal investigator and a research assistant fluent in English and Arabic. Interviews were done in Arabic and then were translated to English?
Response: Information on language of interviews is now presented clearly [Refer to line 117-121].
Comment 6: Were the texts retranslated from English to Arabic?
Response: Interviews conducted in Arabic were transcribed and translated into English. An audit trail of the interviews conducted in Arabic was undertaken by a language expert to check the accuracy of the translation [Refer to line # 121-125].
Comment 7: Were the texts again given to the women to confirm the translated texts from English to Arabic?
Response: Yes, the trustworthiness of the data was assured by undertaking member checks with the study participants. Refugee mothers were contacted to seek verification of the interpretations from the data gathered through multiple sources [Refer to line # 153-155].
Comment 8: Is the language of all Arab countries the same? Do they have different dialects?
Response: Syrian refugee mothers in this study were speaking Arabic, there was no difference in dialects.
Comment 9: Why was the purposeful method not used for the study? For example, based on a quota, the number of years of asylum? Reason for asylum? Motivation for pregnancy? The number of mother's children? Mother's education level? Mother's place of residence? Mother's job and income? Aren't all these things very important in selecting people for interviews?
Response: Yes, both purposive and snowball sampling were used [Refer to line 109-112]. All refugee mothers with young children age 1 day to 24 months who provided consent and agreed to share their breastfeeding experiences were recruited. No restrictions were placed on any other demographic characteristics.
Comment 10: How did the data reach saturation?
Response: In-depth interviews were conducted until data saturation was obtained. Altogether 27 interviews were undertaken. [Refer to line 107-109].
Comment 11: How was the access to the participants? “
Response: Participants were recruited with the help of the refugee settlement organizations offering services to the refugee population in Saskatchewan, Canada [Refer to line 111-112].
Comment 12: To supplement the field observations, a review of media communications ……” How were they chosen?
Response: Reports and websites of service providers focusing on breastfeeding services and resources for refugee mothers were included. [Refer to line 134-139].
Comment 13: Result section: Please design a table and put your data on it.
Response: Table 1 summarizing demographic information is now added.
Comment 14: Please write about creating themes How many codes were obtained? How were the codes classified?
Response: Above information is now added [Refer to line 143-152].
Comment 15: Where were the mothers' countries of origin?
Response: Information is added in the Table 1 and demographic characteristics [Refer to Table 1 and line 171-172]
Comment 16: Please indicate which mother the quotes belong to.
Response: To assure anonymity of participants, as per ethics requirements we have removed names of refugee mothers from the quotes.
Reviewer 4 Report
Comments and Suggestions for Authors
Comments to the Author
Dear Authors,
- Hello, great thanks for reviewing this manuscript.
- The manuscript addresses a topic that is consistent with the scope and aims of the journal.
Abstract
Keywords: Make sure there are keywords inside the mesh.
Introduction:
The introduction is detailed, compact, covering the background information.
But if you can decrease the similarities in plagiarism report in some paragraphs.
Methods:
- The study design and measures are appropriate.
- Need to clearly identify how to get the sample.
- The validity and reliability of the questionnaire was obtained for this study.
Findings:
- Findings are good and clear.
Discussion:
- The Study Discussion provides a clear summary of the main points.

Author Response
Response to the Reviewer 4:
Thanks for the appreciation and valuable feedback. All suggested changes are incorporated.
Comment 1: Keywords: Make sure there are keywords inside the mesh.
Response: We have incorporated this change.
Comment 2: Introduction: The introduction is detailed, compact, covering the background information. But if you can decrease the similarities in plagiarism report in some paragraphs.
Response: We have paraphrased the quotes and have reduced the similarities.
Comment 3: Methods- The study design and measures are appropriate. Need to clearly identify how to get the sample.- The validity and reliability of the questionnaire was obtained for this study.
Response: Information on sample recruitment is added in the methods section [Refer to line 111-113]. As this is a qualitative study, we only used a semi-structed interview guide to undertake in-depth interviews with participants and did not used any questionnaire.
Reviewer 5 Report
Comments and Suggestions for Authors
The study is important. Some suggestions are provided herewith
1. Line 48: expand BFI.
2. Line 49: provide reference/source for there is only one healthcare facility
Material and methods:
3. There is a repetition of content. It needs to be concise (for instance, 'the critical ethnographic study' is repeated several times. 'Principal investigator' is repeated several times).
4. Was the interview conducted in English ???? or the preferred language????. There are contradicting statements (Line 123 and Line 130). Most were in Arabic (line 134). Please specify, how many were in the Arabic language and how many were in the English language.
5. Specify the ethics committee approval number.
6. Include when the study was conducted (eg. Data were collected from ---to ---, giving the duration).
Results:
5. Only 26 mothers are accounted for, in the age distribution. (Line 173-175). That needs to be clarified.
6. Three paragraphs describe the demographic characteristics, which could be made comprehensive by placing a table and describing only the key features.
7. Keywords: Suggest including 'critical ethnography'
8. Overall :
a. Some aspects of the manuscript are presented in the first person ('we/ our) while some others are presented in the second person. It would be appropriate to maintain consistency.
c. Paraphrasing of the content is required (see ithenticate report)
d. Materials and Methods section: Avoid repetition and make it concise.
e. The manuscript requires editing.
Overall this study is of significance and will be of value to the readers.
Comments on the Quality of English Language
As already stated, some aspects of the manuscript will have to be made concise (avoiding repetition of content). The manuscript requires editing.
Author Response
Response to Reviewer: Thanks for the appreciation and valuable feedback. All suggested changes are incorporated in the revised draft of this manuscript.
Comment 1: Line 48: expand BFI
Response: BFI expanded [Refer to line 50-51]
Comment 2: Line 49: provide reference/source for there is only one healthcare facility
Response: Reference added [Refer to line 52].
Comment 3. Material and methods: There is a repetition of content. It needs to be concise (for instance, 'the critical ethnographic study' is repeated several times. 'Principal investigator' is repeated several times).
Response: Repetitive phrases have been removed
Comment 4: Was the interview conducted in English ???? or the preferred language????. There are contradicting statements (Line 123 and Line 130). Most were in Arabic (line 134). Please specify, how many were in the Arabic language and how many were in the English language.
Response: Information on language of interview is now presented clearly [Refer to line 117-124].
Comment 5. Specify the ethics committee approval number
Response: Ethics committee approval number is added [Refer to line 100]. This information is also available in the ethical statement at the end of the manuscript.
Comment 6: Include when the study was conducted (e.g. Data were collected from ---to ---, giving the duration).
Response: Information on time period of data collection is added [Refer to line 100].
Comment 7: Results Only 26 mothers are accounted for, in the age distribution. (Line 173-175). That needs to be clarified.
Response: The information on demographics on age is corrected. [Refer to line 162-165].
Comment 8: Three paragraphs describe the demographic characteristics, which could be made comprehensive by placing a table and describing only the key features.
Response: A table summarizing demographic characteristic is added [Refer to table 1]. Also, only relevant demographic characteristics are retained under the section 3.1 Demographic Characteristics of Study Participants.
Comment 9: Keywords: Suggest including 'critical ethnography'
Response: Keyword ‘critical ethnography’ is added.
Comment 10. Overall: A Some aspects of the manuscript are presented in the first person ('we/ our) while some others are presented in the second person. It would be appropriate to maintain consistency. c. Paraphrasing of the content is required (see ithenticate report) d. Materials and Methods section: Avoid repetition and make it concise. (done) e. The manuscript requires editing.
Response: Manuscript is edited as per the received suggestions. Consistency in language is now assured in the revised draft of this manuscript. Use of “We/our” is avoided. Content is paraphrased. Material and methods section is made concise by removing redundant information.
Round 2
Reviewer 3 Report
Comments and Suggestions for Authors
Thank you for all well revision,